# Composition and Function of Neutrophil Extracellular Traps

**DOI:** 10.3390/biom14040416

**Published:** 2024-03-29

**Authors:** Yijie Wang, Chunjing Du, Yue Zhang, Liuluan Zhu

**Affiliations:** 1Beijing Key Laboratory of Emerging Infectious Diseases, Institute of Infectious Diseases, Beijing Ditan Hospital, Capital Medical University, Beijing 100015, China; 2Beijing Institute of Infectious Diseases, Beijing 100015, China; 3National Center for Infectious Diseases, Beijing Ditan Hospital, Capital Medical University, Beijing 100015, China; 4Department of Critical Care Medicine, Beijing Ditan Hospital, Capital Medical University, Beijing 100015, China

**Keywords:** neutrophil extracellular trap, proteins, host defense, cancer, autoimmune diseases

## Abstract

Neutrophil extracellular traps (NETs) are intricate fibrous structures released by neutrophils in response to specific stimuli. These structures are composed of depolymerized chromatin adorned with histones, granule proteins, and cytosolic proteins. NETs are formed via two distinct pathways known as suicidal NETosis, which involves NADPH oxidase (NOX), and vital NETosis, which is independent of NOX. Certain proteins found within NETs exhibit strong cytotoxic effects against both pathogens and nearby host cells. While NETs play a defensive role against pathogens, they can also contribute to tissue damage and worsen inflammation. Despite extensive research on the pathophysiological role of NETs, less attention has been paid to their components, which form a unique structure containing various proteins that have significant implications in a wide range of diseases. This review aims to elucidate the components of NETs and provide an overview of their impact on host defense against invasive pathogens, autoimmune diseases, and cancer.

## 1. Introduction

Innate immunity constitutes the body’s primary defense against invading pathogens and plays a crucial role in guiding adaptive immunity during persistent infections. Neutrophils, key components of the innate immune system, are among the first responders recruited to sites of inflammation. During the early stages of infection, activated endothelial cells and resident macrophages attract circulating neutrophils, directing them to quickly migrate from the bloodstream to infected tissues, where they efficiently bind, phagocytose, and eliminate microbes [1]. Neutrophils employ three primary mechanisms to directly combat microorganisms: phagocytosis, degranulation, and the generation of neutrophil extracellular traps (NETs). Discovered in 2004 by Brinkmann et al., NETs are structures mainly composed of smooth DNA fibers, as observed via high-resolution scanning electron microscopy [2]. When stimulated with PMA and IL-8, neutrophils release NETs, which contain chromatin DNA, histones, and various proteins such as neutrophil elastase, cathepsin G, myeloperoxidase, lactoferrin, and gelatinase. The process of NET formation, known as “NETosis”, is a type of cell death mechanism distinct from necrosis and apoptosis. While initially thought to occur during active cell death [3], NETosis can also transpire without cell lysis, known as vital NETosis [4]. Notably, Yousefi et al. demonstrated that neutrophils stimulated with granulocyte/macrophage colony-stimulating factor (GM-CSF) followed by Toll-like receptor 4 (TLR4) or complement factor 5a (C5a) receptor activation can produce mitochondrial NETs containing mitochondrial but not nuclear DNA, and are referred to as mitochondrial NETs [5].

While NETs play a vital role in trapping and eliminating pathogens, pathogens have developed strategies to evade immune responses, prompting neutrophils to initiate more aggressive reactions [6]. The excessive activation of NETs and the consequent pro-inflammatory response can result in severe tissue damage [7]. Moreover, beyond their pivotal role in antimicrobial immunity, NETs have been implicated in the pathophysiology of various non-infectious diseases, including autoimmune diseases, atherosclerosis, and cancers, underscoring their dual nature [7]. Despite numerous studies elucidating the mechanisms underlying NET formation, less attention has been paid to their composition. Therefore, this review aims to provide a succinct overview of NET types, formation mechanisms, constituent components, and their implications in diverse diseases.

## 2. Types and Formation Mechanisms of NETs

Three primary forms of NET release have been identified: suicidal NETosis, vital NETosis, and mitochondrial NETosis.

### 2.1. Suicidal NETosis

Suicidal NETosis can be triggered by various stimuli such as bacteria, fungi, viruses, antibody–antigen complexes, autoantibodies, TNF-α, and H_2_O_2_ [8,9,10,11]. This process typically initiates 1–4 h after activation and follows a programmed cell death pathway characterized by nuclear membrane disassembly, chromatin decondensation, binding of depolymerized chromatin to cytoplasmic proteins, and subsequent release of NETs after plasma membrane permeabilization. The stimulation of neutrophils with phorbol 12-myristate 13-acetate (PMA) activates NOX via protein kinase C (PKC) and the rapidly accelerated fibrosarcoma (Raf)-MEK (MAPK/ERK kinase)-ERK signaling pathway [12]. This NOX activation leads to the generation of reactive oxygen species (ROS) and increases calcium influx [13], subsequently activating protein arginine deiminase 4 (PAD4). PAD4 hypercitrullinates histones H3, H2A, and H4, resulting in loss of positive histone charge and chromatin condensation [14]. Concurrently, myeloperoxidase (MPO) and neutrophil elastase (NE) are released from cytoplasmic azurophilic granules, while DNA is liberated from histone degradation by NE and MPO [15]. Following complete cytoskeletal disassembly, the nuclear membrane ruptures, releasing cytoplasmic chromatin that mixes with cytoplasmic proteins [16]. Throughout NETosis, changes in plasma membrane permeability occur, facilitating cell lysis and NET release, aided by Gasdermin D (GSDMD) [17], a pore-forming protein. During PMA-induced NET release, NE cleaves GSDMD to its active form, GSDMD-NT [18], which forms pores in the plasma and granule membranes, thereby promoting the release of NE and other granule contents.

### 2.2. Vital NETosis

Some researchers have described a “vital” form of NET formation, characterized by the release of NETs composed of nuclear DNA via nuclear membrane vesiculation and vesicular export rather than plasma membrane rupture and lysogenic cell death. Vital NETosis differs from PMA-stimulated suicidal NETosis in several aspects, including different stimuli, shorter release times, and independence from NOX [13]. Under septic conditions, platelets activated by TLR4 bind to neutrophils, inducing rapid NET release within a few minutes while limiting the entry of Sytox green, a nucleic acid-staining dye that does not penetrate live cells [19]. In another study, in vivo polymorphonuclear cells rapidly released extracellular traps during skin infection with Gram-positive bacteria [20]. NETs were formed and released very rapidly after 5–60 min of exposure to *Staphylococcus aureus*, independent of ROS production by NOX but mediated by TLR2-dependent mechanisms [8,20]. Unlike suicide NETosis, neutrophils releasing vital NETs retain some normal neutrophil functions such as phagocytosis and migration [20].

### 2.3. Mitochondrial NETosis

In addition to the well-documented formation of NETs with nuclear DNA release, researchers have also observed the presence of NETs containing mitochondrial DNA. Yousef et al. (2009) introduced the concept of mitochondrial NET formation, where NETs consisting of mitochondrial DNA were released after 15–20 min of neutrophil exposure to GM-CSF and lipopolysaccharide (LPS) or C5a [5]. Via live-cell fluorescence microscopy and subsequent quantitative polymerase chain reaction analysis of DNA components, mitochondrial NETs were identified in the blood of patients undergoing skeletal injuries and surgical procedures [21]. Human neutrophils were observed to release mitochondrial DNA from NETs within 5 min of encountering *S. aureus*, a process that relies on mitochondrial complex III [22]. Some researchers classify mitochondrial NETs as vital NETs, as they are released by living cells without inducing lysogenic cell death. The release of mitochondrial NETs, which contain mitochondrial DNA-binding granule proteins, is regulated by mitochondrial ROS and calcium-activated small conductance potassium channels [22,23]. Furthermore, Sirtuin 1 (SIRT1), an enzyme primarily located in the cell nucleus that deacetylates transcription factors, has recently been shown to play a role in tumor-associated aged neutrophils by promoting the opening of mitochondrial permeability transition pore channels for the release of mitochondrial NETs [24]. Notably, Optic atrophy 1 (OPA1), a mitochondrial inner membrane protein involved in ATP production via glycolysis, facilitates the reorganization of the microtubule network and subsequent release of mitochondrial NETs [25].

## 3. Composition of NETs

NETs, regardless of type, consist of DNA along with a variety of conserved nuclear, granular, and cytoplasmic proteins. Key bactericidal proteins found in NETs include histones (H2A, H2B, H3, and H4), granule proteases (NE, MPO, and PR3, etc.), and cytoplasmic proteins like calprotectin and lactoferrin. The presence of antimicrobial proteins such as Azurocidin (AZU1), Cathelicidin, Lysozyme C, and BPIB2 on NETs has also been reported [26]. Initially, 24 proteins were identified in NETs formed by neutrophils stimulated with PMA [27]. Subsequent studies expanded this list, revealing additional proteins induced by PMA stimulation [28]. NETs induced by both mucus and non-mucus strains of *Pseudomonas aeruginosa* shared 33 common proteins and up to 50 variable proteins [26]. However, the compositions of NETs may vary depending on the inducing conditions. Comparison of the NET composition produced by neutrophils under different stimuli, including PMA, calcium ionophore A23187, and *Escherichia coli* LPS, showed divergences between LPS-induced and spontaneous NETs in protein composition, while PMA- and A23187-induced NETs appeared more similar [29]. In total, 330 proteins were identified, 74 of which were detected in all NET preparations.

The composition of NET components can vary depending on the stimulus used to induce NETosis, rather than differences in neutrophil physiology. Different diseases may exhibit varying protein profiles in NETs, with some components playing roles in host defense, cancer, and autoimmune diseases. For example, the dynamin DNAH5 and heat shock protein HSPA1B were prominent in the PMA-induced NETs, while calpain S100A9 and ribosomal protein RPS27 were found in both PMA-induced and A23187-induced NETs. Lysosome-associated membrane protein 2 (LAMP2) was enriched in LPS-induced NETs [29]. Chapman et al. compared the protein composition of NETs from patients with systemic lupus erythematosus (SLE) and rheumatoid arthritis (RA) induced by PMA or A23187; they found minimal differences in NET proteins between the two diseases [30] (Table 1). This indicates that the nature of the stimulus determines the NET protein profile in a certain disease rather than the biological status of neutrophils. Overall, the components and expression levels of proteins in NETs can vary across different diseases, whether infectious or non-infectious (Figure 1).

The components within NETs serve crucial functions in host defense, cancer progression, and autoimmune diseases. In terms of host defense, NETs not only ensnare bacteria and fungi but also exhibit bactericidal properties via their antibacterial constituents. Conversely, in cancer, components such as NE and MMP-9 assume pivotal roles in fostering tumor cell proliferation, metastasis, angiogenesis, and the reactivation of dormant tumor cells. In autoimmune diseases, NET components contribute to the production of autoantibodies, exacerbating autoimmune response. The proteins shown are representative only. Created with BioRender.com. Abbreviations: NE, neutrophil elastase; MPO, myeloperoxidase; CG, Cathepsin G; PR3, myeloblastin; AZU1, azurocidin; MMP-9, matrix metalloproteinase 9; HMGB1, high-mobility group box 1; IFN-α, interferon alpha; TLR-9, Toll-like receptor 9; pDC, plasmacytoid dendritic cells.

## 4. Roles of NET Components in Various Diseases

### 4.1. The Roles of NET Components in Host Defense against Invasive Pathogens

#### 4.1.1. NETs in Bacterial Defense

NETs play a crucial role in the immobilization and elimination of bacteria. Studies employing flow chamber systems or in vivo microscopy have demonstrated NETs’ efficacy in capturing *E. coli* both in vitro and in vivo [19]. Increasing evidence suggests that NETs possess direct antimicrobial properties, effectively combating pathogens such as *E. coli* or *Leishamania* spp. by locally elevating the concentration of antimicrobial agents [59,60]. Notably, chromatin and histones, integral components of NETs, exhibit intrinsic antibacterial activity [61,62]. The DNA within NETs acts as a cation chelator, disrupting both the outer and inner membranes of bacteria like *P. aeruginosa* [31]. Histone H2A stands out as one of the most potent antimicrobial agents, enhancing the pore-forming ability of antimicrobial peptides (AMPs) such as LL-37 (Cathelicidin antimicrobial peptide) and magainin-2 on bacterial membranes [34]. This action depolarizes bacterial membrane potential and impedes membrane recovery. Moreover, H2A’s entry into the cytoplasm of bacteria like *E. coli* and *S. aureus* results in chromosomal reorganization and transcriptional repression [34]. Studies have demonstrated that while bacteria may recover from the pore-forming effects of LL-37, the combined action of H2A and LL-37 is irreversible [34]. In addition, histone H4, isolated from human sebocyte extracts, exhibits antibacterial activity against *S. aureus* and *Propionibacterium* acnes [36]. Histones H2A and H2B have been found to be lethal to *Leishmania* spp., causing surface destruction of histone-treated pro-flagellates, as observed via scanning electron microscopy [35]. Apart from their role in host defense, extracellular histones exhibit cytotoxic effects on various cell types and organs, contributing to septic death by damaging endothelial cells [63,64]. Pentraxin 3 (PTX3) mitigates extracellular histone-mediated cytotoxicity via copolymerization, recognizing pathogens and enhancing the bactericidal efficiency of AZU1 and MPO by binding to them [46,65].

NETs encompass a variety of proteins, including NE, MPO, Cathepsin G, and PR3, all of which possess known antimicrobial properties. NE, for instance, effectively eliminates bacteria in vitro by cleaving proteins on the bacterial outer membrane and targeting virulence factors of pathogens such as *Shigella* or *Yersinia* [42]. Serine proteases like Cathepsin G and PR3 have demonstrated the capability to disrupt virulence factors across various microbial classes [49]. Human neutrophil α-defensin-1 (HNP-1) and lysozyme, for instance, inhibit the production of exotoxins—including hemolysins and superantigens—essential for *S. aureus* infection [50]. Heat shock protein 72 (Hsp70), identified within the DNA of NETs, when released by *M. tuberculosis*-activated neutrophils, stimulates macrophages via TLR2 and TLR4, activating the NF-κB pathway. This activation triggers the release of cytokines such as TNF-α, IL-1β, IL-6, IL-12, and nitric oxide, thereby facilitating the elimination of *M. tuberculosis* by macrophages [51]. Notably, although *S. aureus* and *P. aeruginosa* captured by NETs may survive, the complement proves effective in bacterial elimination. However, NETs hinder the complement’s bactericidal ability, which can be restored by DNase enzyme-mediated NET degradation [66]. This suggests a role for NET-mediated capture in minimizing or preventing bacterial spread until the immune system and DNase enzymes clear NETs via phagocytosis.

#### 4.1.2. NETs in Fungal Defense

NETs serve as a crucial defense mechanism against fungal infections as well. *C. albicans* and *A. fumigatus* can be ensnared by NETs released from human peripheral neutrophils, as well as by NETs present in the lungs of mice [10,67]. Neutrophils employ a microbial size-sensing mechanism to trigger NET formation when encountering large microorganisms like *C. albicans*, which form extensive filamentous hyphae [68]. Antifungal proteins within NETs, including NE, MPO, calprotectin, and lactoferrin, play pivotal roles in host defense [27,69]. Reduced MPO expression in NETs due to mutations in the JAGN1 gene leads to diminished antifungal activity, highlighting MPO’s significance [47]. Studies indicate that NET-enriched supernatants exhibit fungicidal activity against yeasts induced via CAP67, with NE, MPO, collagenase, and histones being crucial components [52]. Lactoferrin, aside from its direct microbicidal activity, interacts with fungal cell surfaces, enhancing cell surface permeability in Candida species and ultimately leading to cell death [53,70]. However, lactoferrin’s charge–charge interaction with NETs may prevent NET formation but lacks the capacity to inhibit excessive NET formation during disease [71]. Calprotectin within NETs sequesters extracellular zinc from *A. fumigatus*, inhibiting conidial germination [54]. Conversely, proteins like Agglutinin-like sequence protein family 3 (Als3) and enolase in *C. albicans* bind to NET proteins, exacerbating tissue damage and accelerating fungal invasion [39]. Pneumocystis pneumonia (PCP) is a fungal lung infection caused by *Pneumocystis jirovecii*. During our investigation into the role of NETs in PCP, our team discovered that *Pneumocystis* induces leukotriene B4-dependent neutrophil swarming, resulting in the formation of agglutinative NETs. However, these agglutinative NETs did not effectively eliminate *Pneumocystis*; instead, they compromised the neutrophils’ ability to kill the fungus [72]. As noted in prior research, the composition of NETs can vary depending on the stimulus. It is hypothesized that the reduced antifungal activity of agglutinative NETs may stem from the absence of essential antifungal components. This phenomenon could represent a strategy employed by *Pneumocystis* to disrupt host innate immune responses and evade immune detection. Investigating the mechanisms underlying the formation of agglutinative NETs and identifying the key molecules responsible for their compromised antifungal response may offer promising targets for the development of antifungal therapies.

#### 4.1.3. NETs in Viral Defense

NETs not only entrap and eliminate bacteria and fungi but also serve a protective role against viral infections, including influenza virus, human immunodeficiency virus (HIV), and respiratory syncytial virus (RSV) [11,73,74]. Our previous studies revealed elevated levels of plasma NETs in patients severely infected with influenza A virus (IAV) and severe fever with thrombocytopenia syndrome (SFTS), suggesting that plasma cfDNA could serve as a useful predictive biomarker of clinical outcome in SFTS and influenza prognosis [75,76,77]. α-Defensin-1 and LL37 exhibit antiviral properties in IAV-induced NET formation. α-Defensin-1 inhibits viral replication by blocking the PKC pathway, while LL37 stimulates NET production in vitro and enhances NET formation upon virus preincubation [78]. Notably, LL-37 increases neutrophil H_2_O_2_ responses to IAV, leading to reduced pro-inflammatory cytokine responses during in vivo IAV infection [41].

Histones play a crucial role in antiviral defense. Histones H3 and H4, rich in arginine, show stronger viral neutralizing and aggregation activity than H2A and H2B, which are rich in lysine [37]. Among core histones, H4 has the highest IAV-neutralizing ability, reducing viral uptake and replication [37]. In HIV-induced NETs, positively charged extracellular histones attract negatively charged HIV-1, preventing its transmission [11]. In addition, MPO and α- Defensin can deactivate HIV-1 by increasing local concentration [11]. However, histones in NETs can also detrimentally affect antiviral activity. Severe acute respiratory syndrome 2 (SARS-CoV-2) causes COVID-19, a highly contagious viral respiratory illness. One of the structural proteins of SARS-CoV-2, spike (S) protein, is responsible for receptor recognition and cell membrane fusion processes [79]. The histones H3 and H4 released by NETosis facilitate membrane fusion and enhance SARS-CoV-2 infectivity by bridging the S2 protein and sialic acid of host cells [38]. In addition to histones, other proteins in NETs, such as NE, PR3, and CG, exhibit antiviral activity against human RSV, even at non-cytotoxic concentrations [45]. NE and PR3 can cleave the viral F protein, hindering virus adhesion and fusion with target cells, further elucidating the antiviral role of NETs [45,80].

### 4.2. The Roles of NET Components in Cancer

NETs emerge as promising diagnostic and prognostic biomarkers in cancer patients, showing associations with the advancement of various malignancies, including gastric, breast, ovarian, hepatocellular, colorectal, and lung cancers [81]. Their induction by factors like tumors or infections contributes to cancer-related proliferation, metastasis, recurrence, and microenvironmental changes. Cools-Lartigue et al. extensively elucidated the involvement of NETs and their component proteins (NE, CG, and matrix metalloproteinase 9 (MMP-9)) in promoting tumor proliferation and dissemination, inhibiting tumor apoptosis, and fostering angiogenesis [82].

Tumors instigate NET formation, thereby facilitating tumor proliferation and dissemination. Linbin Yang and colleagues discovered a transmembrane DNA receptor called CCDC25, which is essential for metastasis facilitated by NETs. This receptor recognizes extracellular NET-DNA, activating the ILK-β-parvin pathway and mediating distant tumor metastasis [33]. NE directly stimulates tumor cell proliferation in human and mouse lung adenocarcinomas by entering the endosomal compartment and degrading insulin receptor substrate-1 [43]. Moreover, Michio Okamoto et al. demonstrated that NE released during NET formation activates extracellular signal-regulated kinase, promoting colorectal cancer cell migration [83].

Disseminated cancer cells often enter a dormant state, remaining inactive until triggered by external stimuli within the microenvironment [84]. In a notable study, NETs induced by tobacco smoke or intranasal administration of LPS were found to awaken dormant breast cancer cells in the lungs, prompting their transition to actively growing metastases [55]. This stimulatory effect was attributed to NET-derived proteases NE and MMP9, which remodel laminin and activate signaling via integrin α3β1, subsequently promoting cancer cell proliferation [55]. Notably, integrin β1 signaling is intricately linked to the transition from dormancy to metastatic proliferation [85]. Laminin is an important component of the extracellular matrix (ECM), and its degradation facilitates the breach of the basement membrane, marking the initial step in tumor cell invasion and metastasis [86]. NETs possess the capacity to degrade the ECM via proteases (e.g., MMP9, NE), releasing vascular endothelial growth factor (VEGF) and platelet-derived growth factors (PFs) to enhance the invasive potential of esophageal cancer cells and facilitate angiogenesis [87]. Calprotectin, typically present at low extracellular levels in NETs, has been found to upregulate the expression of genes such as Cxcl1, Ccl5, Ccl7, Slc39a10, Lcn2, Zc3h12a, and Enpp2 by inducing NF-kB activation, thereby assisting angiogenesis and tumor migration [88].

### 4.3. The Roles of NET Components in Autoimmune Diseases

Autoimmune diseases are characterized by pathological conditions in which the immune system shows intolerance towards self-antigens. Components of NETs can act as damage-associated molecular patterns (DAMPs), contributing to distal organ damage during chronic inflammation [89]. NETs are implicated in inflammation and autoimmunity across various autoimmune diseases, including RA and SLE [90]. Petretto et al. identified a substantial proportion of autoimmune disease-related proteins in the protein composition of NETs using mass spectrometry, consistent with findings by Darrah et al. [29,91].

RA is characterized by the presence of specific autoantibodies against post-translationally modified proteins, such as anti-citrullinated protein antibody (ACPA), in serum and synovial fluid samples from patients [92,93]. Elevated levels of NET components have been observed in peripheral blood, synovial fluids, rheumatoid nodules, and skin of patients with RA, with MPO-DNA complex levels positively correlated with ACPA concentrations [94]. In addition, a variety of monoclonal antibodies found in the synovial fluid and serum of RA patients react with citrullinated proteins (H2A/H2B and fibrinogen) [95]. Citrullinated peptides, generated by histone citrullination via PAD2 and PAD4 activity, are recognized by ACPA, forming immune complexes that induce NET formation. This process leads to the release of neutrophil granule contents as well as cytoplasmic self-antigens from the joints [96].

SLE is a chronic autoimmune disease characterized by the immune system’s attack on healthy tissues throughout the body. Within the blood, there exists a subpopulation of neutrophils known as low-density granulocytes (LDGs), which are immature and exhibit more NET formation, rapid apoptosis, and ROS release upon in vitro stimulation [97]. Patients with SLE often display elevated levels of LDGs in their blood [98]. The presence of NETs and their components in circulation is linked to the production of autoantibodies such as anti-double-stranded DNA (dsDNA), anti-nucleosomal, and anti-histone antibodies, implicated as causative factors in SLE [95]. When NETs are formed by LDG cells, proteins from these structures are released into the microenvironment, including a variety of proteins such as LL37, α- and β-defensins, and high-mobility group box 1 (HMGB1). These proteins bind to DNA and subsequently activate plasmacytoid dendritic cells (pDCs) via Toll-like receptor 9 (TLR9), triggering the synthesis of interferon-alpha (IFN-α) [40,99]. Importantly, IFN-α acts as a potent inducer of NETosis in SLE patients [100], establishing a positive feedback loop that increases the presence of NETs in body fluids. Furthermore, IFN-α promotes the differentiation of autoreactive B cells, leading to the secretion of autoantibodies against cellular components of NETosis and apoptosis [101]. In addition, methyl-oxidized αenolase (methionine sulfoxide 93) plays a significant role in lupus nephritis (LN) and NET formation in patients with SLE, initiating the formation of LN-specific circulating autoantibodies. These autoantibodies are subsequently deposited in the glomeruli, defining LN as the most severe complication of SLE [56].

Anti-neutrophil cytoplasmic antibody (ANCA)-associated vasculitis (AAV) encompasses a group of autoimmune diseases characterized by abnormal infiltration of neutrophils, uncleared leukocyte accumulation in the perivascular tissue, and fibrinoid necrosis of the vessel wall [102]. ANCA often targets autoantigens such as PR3 and MPO, serving as markers of ANCA- AAV [48]. Co-localization of DNA, MPO, and PR3 in renal tissue from patients with small vessel vasculitis (SVV) glomerulonephritis suggests the presence of NET and ANCA antigens in inflamed tissue and MPO-ANCA in serum [48]. However, tissue biopsies of lesions in necrotizing crescentic glomerulonephritis revealed only traces of immune complex deposits, a histological feature that has been termed ‘pauci-immune’ [103]. Inflammatory cytokines like TNF-α stimulate neutrophils to express MPO on their surface, enabling MPO-ANCA to bridge MPO and Fcγ receptors on the cell, thus activating NET formation. NE released during NETosis subsequently digests immunoglobulins in a time- and dose-dependent manner [44]. Additionally, anti-lysosomal membrane protein-2 (LAMP-2) is also an ANCA autoantibody target. Studies by S Tang et al. have shown that anti-LAMP-2 antibodies inhibited spontaneous apoptosis and activated neutrophil autophagy, exacerbating AAV progression by inducing neutrophil NET release in vitro [58].

## 5. Concluding Remarks

Since the initial discovery of NETs, significant advancements have been made in their study, offering novel insights into disease development and therapeutic strategies. Researchers have made considerable progress in understanding the formation mechanisms and composition of NETs via laboratory investigations and clinical observations, revealing their pivotal role in various diseases, including infectious diseases, cancer, and autoimmune disorders. However, controversies persist regarding the precise mechanisms underlying NET formation, and our understanding of the degradation mechanisms of NETs remains incomplete. Additionally, accurately detecting and quantifying NETs presents a significant challenge due to the complexity and diversity of their morphology and composition. While the study of NETs has opened new avenues for understanding disease pathogenesis and treatment, addressing these challenges in practical applications requires further research and exploration.

## Figures and Tables

**Figure 1 biomolecules-14-00416-f001:**
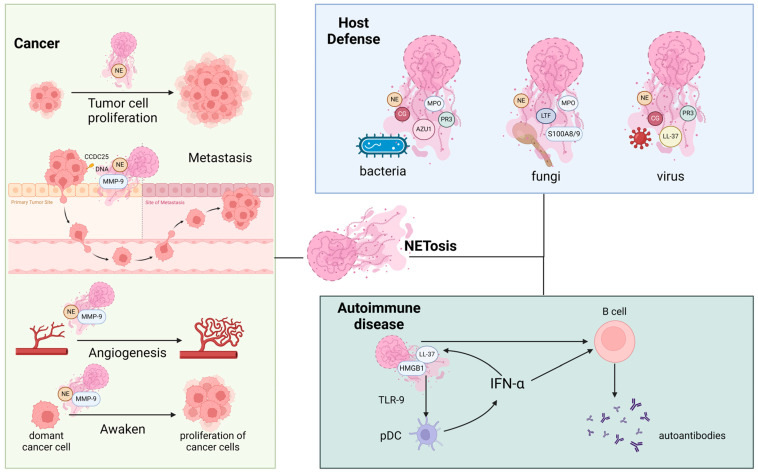
Diverse roles of NETs and their components in various pathological states.

**Table 1 biomolecules-14-00416-t001:** Summary of NET-related components.

NET Component (Protein IDs)	Stimulus	Cellular Localization	Related Diseases or Pathogens	Reference
DNA	Spontaneous, PMA, LPS, A23187	nucleus	*P. aeruginosa*	Mulcahy et al., 2008 Mulcahy, Charron-Mazenod, and Lewenza [31]
			SLE	Yaniv et al., 2015 Yaniv, Twig, Shor, Furer, Sherer, Mozes, Komisar, Slonimsky, Klang, Lotan, Welt, Marai, Shina, Amital, and Shoenfeld [32]
			liver metastases	Yaniv et al., 2015 Yang, Liu, Zhang, Liu, Zhou, Chen, Huang, Li, Li, Chen, Liu, Xing, Chen, Su, and Song [33]
Histone H2A (Q99878)	Spontaneous, PMA, LPS, A23187	nucleus	*E. coli* and *S. aureus*	Doolin et al., 2020 Doolin, Amir, Duong, Rosenzweig, Urban, Bosch, Pol, Gross, and Siryaporn [34]
			*Leishmania* spp.	Wang et al., 2011 Wang, Chen, Xin, Beverley, Carlsen, Popov, Chang, Wang, and Soong [35]
Histones H2B (Q16778)	Spontaneous, PMA, LPS, A23187	nucleus	*Leishmania* spp.	Wang et al., 2011 Wang, Chen, Xin, Beverley, Carlsen, Popov, Chang, Wang, and Soong [35]
Histone H4 (P62805)	Spontaneous, PMA, LPS, A23187	nucleus	*S. aureus* and *Propionibacterium* acnes	Lee et al., 2009 Lee, Huang, Nakatsuji, Thiboutot, Kang, Monestier, and Gallo [36]
			IAV	Hoeksema et al., 2015 Hoeksema, Tripathi, White, Qi, Taubenberger, van Eijk, Haagsman, and Hartshorn [37]
			SARS-CoV-2	Hong et al., 2022 Hong, Yang, Zou, Bi, He, Lei, He, Li, Alu, Ren, Wang, Jiang, Zhong, Jia, Yang, Yu, Huang, Yang, Zhou, Zhao, Kuang, Wang, Wang, Chen, Luo, Zhang, Lu, Chen, Que, He, Sun, Wang, Shen, Lu, Zhao, Yang, Yang, Wang, Li, Song, Dai, Chen, Geng, Gou, Chen, Dong, Peng, Huang, Qian, Cheng, Fan, Wei, Su, Tong, Lu, Peng, and Wei [38]
LL-37 (J3KNB4)	Spontaneous, PMA, LPS, A23187	extracellular	*E. coli* and *S. aureus*	Doolin et al., 2020 Doolin, Amir, Duong, Rosenzweig, Urban, Bosch, Pol, Gross, and Siryaporn [34]
			*C. albicans*	Karkowska-Kuleta et al., 2021 Karkowska-Kuleta, Smolarz, Seweryn-Ozog, Satala, Zawrotniak, Wronowska, Bochenska, Kozik, Nobbs, Gogol, and Rapala-Kozik [39]
			SLE	Lande et al., 2011 Lande, Ganguly, Facchinetti, Frasca, Conrad, Gregorio, Meller, Chamilos, Sebasigari, Riccieri, Bassett, Amuro, Fukuhara, Ito, Liu, and Gilliet [40]
			IAV	Tripathi et al., 2014 Tripathi, Verma, Kim, White, and Hartshorn [41]
neutrophil elastase (P08246)	PMA, LPS, A23187	organelle	*Shigella* or *Yersinia*	Weinrauch et al., 2002 Weinrauch, Drujan, Shapiro, Weiss, and Zychlinsky [42]
			*C. albicans*	Karkowska-Kuleta et al., 2021 Karkowska-Kuleta, Smolarz, Seweryn-Ozog, Satala, Zawrotniak, Wronowska, Bochenska, Kozik, Nobbs, Gogol, and Rapala-Kozik [39]
			lung adenocarcinomas	Houghton et al., 2010 Houghton, Rzymkiewicz, Ji, Gregory, Egea, Metz, Stolz, Land, Marconcini, Kliment, Jenkins, Beaulieu, Mouded, Frank, Wong, and Shapiro [43]
			AVV	Futamata et al., 2018 Futamata, Masuda, Nishibata, Tanaka, Tomaru, and Ishizu [44]
			RSV	Lopes et al., 2022 Lopes, da Silva, de Lima Menezes, de Oliveira, Watanabe, Porto, da Silva, and Toledo [45]
Myeloperoxidase (P05164-2)	Spontaneous, PMA, LPS, A23187	organelle	bacteria	Daigo et al., 2012 Daigo and Hamakubo [46]
			*C. albicans*	Khandagale et al., 2018 Khandagale, Lazzaretto, Carlsson, Sundin, Shafeeq, Römling, and Fadeel [47]
			AVV	Kessenbrock et al., 2009 Kessenbrock, Krumbholz, Schönermarck, Back, Gross, Werb, Gröne, Brinkmann, and Jenne [48]
			HIV-1	Saitoh et al., 2012 Saitoh, Komano, Saitoh, Misawa, Takahama, Kozaki, Uehata, Iwasaki, Omori, Yamaoka, Yamamoto, and Akira [11]
Azurocidin (P20160)	Spontaneous, PMA, LPS, A23187	cytoplasm	host defense	Daigo et al., 2012 Daigo and Hamakubo [46]
Cathepsin G (P08311)	Spontaneous, PMA, LPS, A23187	nucleus, membrane, cytoplasm, extracellular	host defense	Averhoff et al., 2008 Averhoff, Kolbe, Zychlinsky, and Weinrauch [49]
			RSV	Lopes et al., 2022 Lopes, da Silva, de Lima Menezes, de Oliveira, Watanabe, Porto, da Silva, and Toledo [45]
Myeloblastin (PR3) (P24158)	PMA, LPS, A23187	membrane, extracellular	host defense	Averhoff et al., 2008 Averhoff, Kolbe, Zychlinsky, and Weinrauch [49]
			AVV	Kessenbrock et al., 2009 Kessenbrock, Krumbholz, Schönermarck, Back, Gross, Werb, Gröne, Brinkmann, and Jenne [48]
			RSV	Lopes et al., 2022 Lopes, da Silva, de Lima Menezes, de Oliveira, Watanabe, Porto, da Silva, and Toledo [45]
Neutrophil defensing (P59665)	PMA	extracellular	*S. aureus*	Schlievert et al., 2023 Schlievert, Kilgore, Beck, Yoshida, Klingelhutz, and Leung [50]
			SLE	Lande et al., 2011 Lande, Ganguly, Facchinetti, Frasca, Conrad, Gregorio, Meller, Chamilos, Sebasigari, Riccieri, Bassett, Amuro, Fukuhara, Ito, Liu, and Gilliet [40]
			IAV	Tripathi et al., 2014 Tripathi, Verma, Kim, White, and Hartshorn [41]
			HIV-1	Saitoh et al., 2012 Saitoh, Komano, Saitoh, Misawa, Takahama, Kozaki, Uehata, Iwasaki, Omori, Yamaoka, Yamamoto, and Akira [11]
Heat shock protein 72 (P08107)	Spontaneous, PMA, LPS, A23187	nucleus, cytoplasm, extracellular	*M. tuberculosis*	Braian et al., 2013, Braian, Hogea and Stendahl [51]
Interstitial collagenase (P03956)	-	extracellular	fungi	Rocha et al., 2015 Rocha, Nascimento, Decote-Ricardo, Côrte-Real, Morrot, Heise, Nunes, Previato, Mendonça-Previato, DosReis, Saraiva, and Freire-de-Lima [52]
Lactotransferrin (P02788)	Spontaneous, PMA, LPS, A23187	cytoplasm, extracellular	*Candida* species’	Nikawa et al., 1993 Nikawa, Samaranayake, Tenovuo, Pang, and Hamada [53]
Calprotectin S100A8(P05109) S100A9(P06702)	Spontaneous, PMA, LPS, A23187	membrane, cytoplasm, extracellular	*C. albicans*	Urban et al., 2006 Urban, Reichard, Brinkmann, and Zychlinsky [10]
			*A. fumigatus*	Gazendam et al., 2016 Gazendam, van Hamme, Tool, Hoogenboezem, van den Berg, Prins, Vitkov, van de Veerdonk, van den Berg, Roos, and Kuijpers [54]
matrix metalloproteinase 9 (P14780)	LPS	extracellular	cancer	Albrengues et al., 2018 Albrengues, Shields, Ng, Park, Ambrico, Poindexter, Upadhyay, Uyeminami, Pommier, Küttner, Bružas, Maiorino, Bautista, Carmona, Gimotty, Fearon, Chang, Lyons, Pinkerton, Trotman, Goldberg, Yeh, and Egeblad [55]
α-enolase (P06733)	Spontaneous, PMA, LPS	cytoplasm	SLE	Bruschi et al., 2019 Bruschi, Petretto, Santucci, Vaglio, Pratesi, Migliorini, Bertelli, Lavarello, Bartolucci, Candiano, Prunotto, and Ghiggeri [56]
Annexin A1 (P04083)	Spontaneous, PMA, LPS, A23187	cytoplasm	SLE	Bruschi et al., 2019 Bruschi, Petretto, Santucci, Vaglio, Pratesi, Migliorini, Bertelli, Lavarello, Bartolucci, Candiano, Prunotto, and Ghiggeri [56]
High-mobility group box 1 (P09429)	Spontaneous, PMA, LPS, A23187	nucleus, cytoplasm	SLE	Whittall-García et al., 2019 Whittall-García, Torres-Ruiz, Zentella-Dehesa, Tapia-Rodríguez, Alcocer-Varela, Mendez-Huerta, and Gómez-Martín [57]
lysosomal membrane protein-2 (P13473)	-	membrane	AVV	Tang et al., 2015 Tang, Zhang, Yin, Gao, Shi, Wang, Huang, Wang, Zou, Zhao, Huang, Shan, Gounni, Wu, and Zhang [58]

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
