# Peer review of "Composition and Function of Neutrophil Extracellular Traps"

_biomolecules, 2024, doi:10.3390/biom14040416_

Round 1

Reviewer 1 Report

Comments and Suggestions for Authors

The authors have done a great job in which, unlike other reviews on the subject, which focus fundamentally on the types of NETosis, they have analyzed the composition of NETs depending on whether we are dealing with suicidal, non-suicidal or mitochondrial NETosis. Furthermore, they evaluate the composition in NETs generated in response to bacterial, fungal infections, cancer or autoimmune diseases.

In my opinion, the roles of NET components in host defense against viruses should be included

Reviewer 2 Report

Comments and Suggestions for Authors

The review article entitled “Composition and Function of Neutrophil Extracellular Traps” by Wang et al is outstanding. It focuses primarily on the proteins known to be acquired by NETs and which serve a number of bioactive roles in infection, cancer and autoimmunity, all of which are covered by the authors. Also, the focus on associated NET proteins separate this review from the many other reviews that concentrate on other aspects of NETosis. The paper is very well written and I saw no grammatical or spelling errors which reflects careful preparation. I can identify no weaknesses.

Reviewer 3 Report

Comments and Suggestions for Authors

The manuscript gives a well structured summary of current knowledge of NETs and NETosis, focusing on the component proteins of NETs and how they vary with induction modes. Further it recounts the roles of NETs components in host defense, cancer and autoimmune diseases. Thus it provides a welcome summary of this extensive topic and emphasize the importance of future research focusing on distinct NET components. The manuscript is well structured, and the English language is free of grammatical errors and mostly phrased clearly. The references include most current and relevant articles in the filed.

Minor points:

Lines 131-133

‘This evolution was made possible through the use of the High Throughput Fusion Orbitrap, a new technology that offers much higher sensitivity compared to previous mass spectrometry methods’

This sentence is not relevant and should be omitted.

Line 143

‘they found minimal differences in NET proteins’

The use of the word minimal suggests that this finding is not significant.

Lines 179/184

In my opinion, no new paragraph is needed after ‘…. as observed via scanning electron microscopy’ but there should be a new paragraph after ‘…. recognizing pathogens and enhancing the bactericidal efficiency of AZU1 and MPO by binding to them.’

Lines 202 – 204

‘C. albicans and A. fumigatus are ensnared by NETs released from human neutrophils, evident in infected mice lungs’

This sentence would benefit from rephrasing. In my understanding it suggests that human neutrophils are present in infected mice lungs.

Lines 242,243

‘Linbin Yang et al. identified a transmembrane DNA receptor, CCDC25, which relies on NET-dependent metastasis’

This sentence is unclear and would benefit from rephrasing.

Line 250

replace NET by NETs

Line 256

Omit the word ‘meanwhile’

Lines 286,287

‚Within the blood, there exists a subpopulation of neutrophils in the blood known as low-density granulocytes ‘

Omit the second ‘in the blood’

Line 288

‘… exhibit rapid apoptosis and ROS release upon in vitro stimulation’

It should be mentioned there that LDGs also show enhanced NET formation (eg. Herteman et al. Scientific Reports. 2017;7: 7743. doi:10.1038/s41598-017-08089-5)

Line 292

‘When NETs are formed in LDG cells’

Replace ‘in’ by ‘by’

Line 312

‘glomerulonephritis revealed only trace immune complex deposits‘

Should be ‘… traces of immune complex deposits’

Lines 318,319

‘anti-LAMP-2 antibody inhibit spontaneous apoptosis’ should be ‘anti-LAMP-2 antibodies inhibited spontaneous apoptosis’

In Refs. 17, 24, 37 and 51 the pages/article IDs are missing

Comments on the Quality of English Language

The manuscript is well structured, and the English language is free of grammatical errors and mostly phrased clearly. Some minor edits would improve comprehensibility (see above/pdf file).
